# Intestinal Permeability Is a Mechanical Rheostat in the Pathogenesis of Liver Cirrhosis

**DOI:** 10.3390/ijms22136921

**Published:** 2021-06-28

**Authors:** Norihisa Nishimura, Kosuke Kaji, Koh Kitagawa, Yasuhiko Sawada, Masanori Furukawa, Takahiro Ozutsumi, Yukihisa Fujinaga, Yuki Tsuji, Hiroaki Takaya, Hideto Kawaratani, Kei Moriya, Tadashi Namisaki, Takemi Akahane, Hiroshi Fukui, Hitoshi Yoshiji

**Affiliations:** Department of Gastroenterology, Nara Medical University, 840 Shijo-cho, Kashihara, Nara 634-8522, Japan; kajik@naramed-u.ac.jp (K.K.); kitagawa@naramed-u.ac.jp (K.K.); yasuhiko@naramed-u.ac.jp (Y.S.); furukawa@naramed-u.ac.jp (M.F.); ozutaka@naramed-u.ac.jp (T.O.); fujinaga@naramed-u.ac.jp (Y.F.); tsujih@naramed-u.ac.jp (Y.T.); htky@naramed-u.ac.jp (H.T.); kawara@naramed-u.ac.jp (H.K.); moriyak@naramed-u.ac.jp (K.M.); tadashin@naramed-u.ac.jp (T.N.); stakemi@naramed-u.ac.jp (T.A.); hfukui@naramed-u.ac.jp (H.F.); yoshijih@naramed-u.ac.jp (H.Y.)

**Keywords:** leaky gut, endotoxins, alcoholic liver disease, nonalcoholic steatohepatitis, liver cirrhosis, hepatocarcinogenesis, Toll-like receptor 4 pathway

## Abstract

Recent studies have suggested that an alteration in the gut microbiota and their products, particularly endotoxins derived from Gram-negative bacteria, may play a major role in the pathogenesis of liver diseases. Gut dysbiosis caused by a high-fat diet and alcohol consumption induces increased intestinal permeability, which means higher translocation of bacteria and their products and components, including endotoxins, the so-called “leaky gut”. Clinical studies have found that plasma endotoxin levels are elevated in patients with chronic liver diseases, including alcoholic liver disease and nonalcoholic liver disease. A decrease in commensal nonpathogenic bacteria including *Ruminococaceae* and Lactobacillus and an overgrowth of pathogenic bacteria such as *Bacteroidaceae* and *Enterobacteriaceae* are observed in cirrhotic patients. The decreased diversity of the gut microbiota in cirrhotic patients before liver transplantation is also related to a higher incidence of post-transplant infections and cognitive impairment. The exposure to endotoxins activates macrophages via Toll-like receptor 4 (TLR4), leading to a greater production of proinflammatory cytokines and chemokines including tumor necrosis factor-alpha, interleukin (IL)-6, and IL-8, which play key roles in the progression of liver diseases. TLR4 is a major receptor activated by the binding of endotoxins in macrophages, and its downstream signal induces proinflammatory cytokines. The expression of TLR4 is also observed in nonimmune cells in the liver, such as hepatic stellate cells, which play a crucial role in the progression of liver fibrosis that develops into hepatocarcinogenesis, suggesting the importance of the interaction between endotoxemia and TLR4 signaling as a target for preventing liver disease progression. In this review, we summarize the findings for the role of gut-derived endotoxemia underlying the progression of liver pathogenesis.

## 1. Introduction

The interaction between the intestine and liver via microbiologic features from the gut, referred to as the “gut–liver axis”, has been recently demonstrated as contributing to the development of liver diseases that develop liver cirrhosis and hepatocellular carcinoma (HCC). Although there are many mechanisms underlying the pathogenesis of liver inflammation, fibrosis, and carcinogenesis, microbial metabolites and products derived from the intestinal tract are considered to be some of the major factors that accelerate the progression of liver diseases due to the close links between the liver and intestine.

In patients with liver diseases, the gut bacteria themselves, as well as their components, often translocate into the portal blood flow and directly reach the liver as a result of the disrupted intestinal barrier, or so-called bacterial translocation. In chronic liver disease (CLD) patients, dysbiosis is often observed and increases alongside the stages of liver fibrosis, leading to bacterial translocation including endotoxins. Endotoxins (lipopolysaccharides; LPS) are a well-known component of Gram-negative bacteria and work as pathogen-associated molecular patterns (PAMPs) for Toll-like receptor 4 (TLR4), which recognizes unique structural components of bacteria and drives innate immunity-producing inflammatory cytokines, such as tumor necrosis factor-α (TNF-α) and interleukin (IL)-6 [1]. In the liver, Kupffer cells (KCs) are regional macrophages that control the innate immune response and are activated by LPS binding, but LPS also stimulate hepatic nonimmune cells, including hepatic stellate cells (HSCs) and sinusoidal endothelial cells as they also have TLR4 [2,3,4]. In this review, we summarize the influence of gut-derived endotoxins on the progression of liver disease.

## 2. Mechanisms of Endotoxemia Derived from Gut Microbiota

### 2.1. Dysbiosis

Recent studies have demonstrated that the alteration in the gut microbiome referred to as “dysbiosis” is closely associated with the clinical stages of CLDs, including liver cirrhosis, the end stage of chronic hepatitis [5,6,7]. The role of dysbiosis in patients with CLDs has been reported by a study investigating the alteration in gut microbial diversity by 16S sequencing and a meta-analysis in some diseases such as liver cirrhosis, colorectal cancer, inflammatory bowel disease, obesity, and type 2 diabetes mellitus [5]. This study also suggested that liver fibrosis, the F4 stage of fibrosis, is the condition in which dysbiosis is linked to the pathogenesis, including in patients with nonalcoholic steatohepatitis (NASH) [8,9]. Likewise, an alteration in the diversity of the gut microbiota showed the most significant difference between healthy control and cirrhotic patients [5,6]. In patients with hepatitis B (HBV), the proportion of *Bifidobacteria* and *Bacteroidetes* decreased, whereas that of *Enterococcus*, *Enterobacteriaceae*, and *Proteobacteria* increased [10,11,12]. In patients with NASH and diabetes, obesity triggers dysbiosis, which leads to a decrease in the diversity of the gut microbiota and an increased ratio of *Firmicutes* to *Bacteroidetes* [13]. A decrease in *Bacteroidetes* has also been observed in the intestine of alcoholic patients [14]. Furthermore, patients with alcoholic liver disease demonstrated an increase in *Veillonellaceae**,* which has been reported to induce highly systemic inflammation [15]. In Table 1, we list representative alterations in bacterial diversity according to types of CLDs. The evidence from these studies suggests that dysbiosis affects multiple cascades related to the progression of CLD, such as inflammation, fibrogenesis, regeneration, and immunity, leading to the development of HCC.

### 2.2. Small Intestinal Bacterial Overgrowth (SIBO)

Small intestinal bacterial overgrowth (SIBO) is a part of dysbiosis that occurs in the small intestine. It is characterized by an abnormal composition of microbiota in the small intestinal tract [16]. Recently, it has become known to be associated with various diseases and not just liver diseases. SIBO is considered as an important phenomenon that leads to an increased intestinal permeability via the impairment of tight junctions (TJs) by intestinal epithelial cell damage [17,18]. Since the intestinal overgrowth of *Escherichia coli* in patients with liver cirrhosis was reported in 2016, a recent study was the first to demonstrate the change in gut microbial diversity in cirrhotic and HCC patients and its possibility as a novel, noninvasive surrogate marker for the detection of HCC in patients with HBV infection [19,20].

An alteration of bile acid secretion is a major factor in inducing bacterial overgrowth. In patients with liver cirrhosis, as the secretion of bile acid from the liver is decreased, bacterial overgrowth and the change in the bacterial composition are promoted. It has further been demonstrated that the gastrointestinal transition time is longer in cirrhotic patients than in healthy controls because of the dysfunction of intestinal motility, which also leads to the promotion of bacterial overgrowth [21]. Another study reported an association between proximal small intestinal motility and SIBO in patients with liver cirrhosis [22]. As gastric acid affects the regulation of microbiota and prevents the translocation of oral bacteria through the stomach, the inhibition of gastric acid production with the use of a proton pump inhibitor could also be implicated in the development of SIBO [23].

### 2.3. Disruptions in Intestinal Barrier Function

The intestinal barrier is composed of three different components, which are physical, immunologic, and microbial barriers. The physical barrier contains a mucus layer and epithelial elements. The epithelium of the intestine has occlusive intracellular junctions, which are the so-called tight junctions (TJs) [24]. TJs are composed of three types of transmembrane proteins associated with scaffolding proteins that link to the actin cytoskeleton such as occludin, claudins, and junctional adhesion molecules [25]. In particular, claudins are the main molecules within the TJ structure and produce charge-selective pores that regulate the transportation of ions and small particles across the epithelial layers. These proteins bind scaffolding proteins such as zonula occludins 1 (ZO-1), ZO-2, and ZO-3, which connect transmembrane proteins including claudins and occludins to the actin cytoskeleton [26,27,28]. These proteins protect the translocation of bacterial components and their metabolites.

The causes of intestinal permeability vary and remain to be elucidated. Physical damage to the intestinal epithelium, disruption to the TJ, and alterations in the thickness of the mucus layer can result in increased intestinal leakiness [29]. Dysfunctions in either the innate or adaptive immune response can affect the increase in microbe translocation [30]. Dysbiosis can also promote inflammation in the intestinal epithelium and barrier dysfunction [31]. The increased PAMPs such as LPS, bacterial RNAs, and viral RNAs translocate from the intestine through the intestinal barrier and enter the liver, and they stimulate immune cells such as KCs via binding to TLR4 on their membrane, resulting in the induction of hepatic inflammation that progresses to liver injury and diseases (Figure 1) [31,32].

## 3. The Role of Endotoxemia in the Progression of Liver Pathogenesis

### 3.1. Alcoholic Liver Disease

Excessive consumption of alcohol is an important cause of chronic hepatitis that develops into cirrhosis and the occurrence of HCC based on cirrhosis [33,34]. Continuous alcohol intake promotes bacterial overgrowth in both the small and large intestines [35,36,37]. Moreover, drinking alcohol induces bacterial translocation into the portal vein, which plays a crucial role in alcoholic liver injury [38]. A previous study reported that alcohol-induced cirrhotic patients have SIBO [34]. In addition, in these patients, the serum endotoxin levels are elevated even at the early stage of alcoholic liver disease [39]. Leclercq et al. indicated that almost half of the patients with alcohol dependence have increased intestinal permeability, even at the early stage of liver fibrosis (F0–1) [40]. Those authors also demonstrated that the total amount of gut bacteria, especially *Faecalibacterium prausnitzii* in the *Ruminococcaceae* family, was negatively correlated with intestinal permeability, whereas the genera *Dorea* and *Blautia* were positively correlated with intestinal permeability. Kakiyama et al. also reported that *Bacteroidaceae* and *Porphyromonadaceae* were decreased and *Veillonellaceae* was increased in patients with alcoholic liver cirrhosis as compared with those with cirrhosis due to other etiologies [15]. In particular, *Veillonellaceae* could lead to highly systemic inflammation and endotoxemia [15]. The evidence from these studies suggests that dysbiosis is closely associated with increased intestinal permeability and portal endotoxemia.

Endotoxemia derived from the Gram-negative bacteria of the *Proteobacteria* phylum is considered to be a driver of increased hepatic inflammation. Dysbiosis, namely, a decrease in commensal bacteria, induces a disruption in the intestinal TJ barrier [41]. Experimentally, supplementation of the probiotic *Lactobacillus rhamnosus GG* improved downregulated TJ protein expression and attenuated endotoxemia in alcohol-fed mice [42]. This finding could corroborate that improvement of dysbiosis contributes to preventing the disruption of the gut barrier function.

Gut-derived endotoxins from gut bacteria play a crucial role in the pathogenesis of alcoholic liver injury [43,44]. When endotoxins enter into the liver, they initially activate KCs, which are regional macrophages in the liver, via binding of TLR4 on their membrane, leading to an induction of inflammatory cytokines and chemokines including TNF-α, IL-1, and IL-6, mediated by the activation of the nuclear factor–κB pathway (Figure 2) [45]. In addition, alcohol induces the LPS binding protein, which is closely related to the effect of endotoxins. LPS–TLR4 binding on the surface of KCs also induces reactive oxygen species (ROS) production, which in turn induces migration of T lymphocytes and neutrophils and HSC activation [1,2]. Likewise, because nonimmune cells such as HSCs and liver sinusoidal endothelial cells have TLR4, the exposure of endotoxins to their cells further promotes fibrotic activity via the activation of these cells [4].

### 3.2. NASH

Nonalcoholic fatty liver disease (NAFLD) is considered to be the manifestation of metabolic syndrome in the liver. Recently, it has been suggested that the definition and term NAFLD itself should be changed to “metabolic-associated fatty liver disease” [46]. Various studies have indicated that the gut microbiota are closely associated with the pathogenesis of the development of NASH [47].

As compared with lean NAFLD individuals, NAFLD patients with obesity have a tendency to have SIBO and leaky gut. An initial preliminary study demonstrated that a greater population of *Firmicutes* and a smaller population of *Bacteroidetes* were observed in patients with obesity as compared with healthy controls [48]. It was then established that the *Firmicutes*/*Bacteroidetes* ratio was increased in the obese population in both rodents and humans [49,50]. A greater *Firmicutes*/*Bacteroidetes* ratio and increased *Proteobacteria* were associated with negative health effects, such as induction of systemic inflammatory activity by the increased gut permeability [51]. Miele et al. first reported that the expression of the TJ protein ZO-1 was decreased in patients with NAFLD as compared with healthy subjects [52]. A higher serum level of endotoxins was observed not only in adult NASH patients but also in children with NASH as compared with those without NASH [53,54,55]. Patients with NASH also displayed a higher expression level of TLR4 in the liver than in those without NASH [56]. The involvement of TLR4 signaling has also been reported in a study demonstrating the relationship between TLR4 mutation and NAFLD [55]. An experimental study using obese mice demonstrated that increased intestinal permeability via the downregulation of occludins and ZO-1 on the intestinal epithelium is closely associated with portal endotoxemia as well as the elevation of serum inflammatory cytokine levels in obese mice [57]. In obese rats, the liver has an elevated sensitivity against LPS exposure and a reduction in the phagocytic function of KCs [58]. Moreover, a high-fat diet also promotes the translocation of bacterial products including living bacteria through the intestinal mucosa, suggesting the importance of the gut–liver axis in liver injury in NAFLD patients [59].

To date, various studies have already demonstrated the importance of the gut–liver axis and the role of TLR4 signaling on the progression of NAFLD. It was previously reported that mice fed a methionine choline-deficient diet demonstrated steatohepatitis, portal endotoxemia, and elevation of TLR4 expression in the liver, whereas TLR4 mutant mice showed less tissue damage and lipid accumulation in the liver [60]. Our study demonstrated that rats fed a choline-deficient L-amino acid-defined diet, which mimics the NASH liver, showed greater α-smooth muscle actin expression and enhanced LPS binding protein mRNA levels in the liver tissue, increased intestinal permeability, and decreased TJ protein expression in the intestine [61]. In contrast, oral medication with poorly absorbable antibiotics inhibited LPS–TLR4 signaling and suppressed the progression of liver fibrosis [61]. This evidence supports the idea that the prevention of LPS stimulation could be a useful therapeutic strategy for patients with CLD.

### 3.3. Viral Hepatitis

Hepatitis A and E viral infections cause acute hepatitis, which can be self-clearing. Both A and E hepatitis viruses are transmitted through the mouth into the gastrointestinal tract and may affect the diversity of the gut microbiota. Although supplementation of beneficial bacteria such as *Enterococcus faecium* was shown to contribute to the removal of the hepatitis E virus from the intestine of pigs, this effect remains unclear in humans [62].

In contrast, hepatitis B and C viruses are major viruses that cause chronic hepatitis. It is also observed that patients with chronic hepatitis have higher bacterial translocation from the gut [63]. It was recently reported that commensal microbiota play an important role in both the viral host cell interaction and viral replication. Some bacterial species including *Neisseria*, *E. coli*, *Enterobacteriaceae*, *E. faecalis*, *F. prausnitzii*, and *Gemella* have been identified as having a responsible role in the progression of hepatitis B and C infection [64,65].

#### 3.3.1. Hepatitis B Viral Infection

Dysbiosis can influence the progression of disease pathogenesis, resulting in liver failure. In patients with HBV, bacteria producing LPS are enriched. It has been demonstrated that the proportions of *F. prausnitzii*, *E. faecalis*, *Enterobacteriaceae*, *Bifidobacteria*, and *Lactobacillus* were markedly changed in HBV cirrhotic patients [66]. Oral dysbiosis has also been reported during HBV infection, as a reduction in *Bacteroidetes* and an increase in *Proteobacteria*. Another study suggested the positive correlation of *Neisseriaceae* with the level of serum HBV DNA [67].

In contrast, Lu et al. reported that cirrhotic HBV patients exhibited a marked decrease in the ratio of *Bifidobacteriaceae* to *Enterobacteriaceae*. Another study of cirrhotic patients with HBV also found a decrease in *Bifidobacteria* and *Lactobacillus*, whereas the levels of *Enterococcus* and *Enterobacteriaceae* were significantly increased as compared with the healthy population.

#### 3.3.2. Hepatitis C Viral Infection

In most patients with hepatitis C virus (HCV), the amount of *Enterobacteriaceae* and *Bacterioidetes* increased, whereas that of *Firmicutes* decreased. Previous studies found that plasma levels of LPS in HCV patients are elevated due to the promotion of bacterial translocation and intestinal inflammation [68,69]. As the production of bile acids is important for the gut microbial composition [70], several pathogenic bacteria including *Enterobacteriaceae*, *Enterococcus,* and *Staphylococcus* prevent the production of bile acid in patients with HCV, and the use of oral direct-acting antivirals reversed the alteration in the gut microbiota. That study further revealed that oral direct-acting antivirals could be helpful in improving the gut microflora through a reduction in intestinal inflammation via mediating the TNF-α level [71].

Several beneficial bacteria have been reported to have a protective effect during HCV infection. These bacteria including *Bifidobacterium* spp. and *L. acidophilus* play a supportive role in the antiviral effect, even in antibacterial activities [72]. The immune response by natural killer cells can be extended by protective bacteria such as probiotics in HCV-infected patients, and they also promote the cytotoxic effect of natural killer cells to inhibit HCV replication [73,74].

### 3.4. Autoimmune Liver Diseases

Primary biliary cholangitis (PBC) is one of the common autoimmune liver diseases characterized by hepatobiliary injury, which shows progressive nonsuppurative destruction of small intrahepatic bile ducts, resulting in cholestasis, inflammation, and fibrosis. An increased diversity of *Eubacterium* and *Veillonella* and a decreased diversity of *Fusobacterium* in the oral microbiota were demonstrated in Japanese patients with PBC as compared with healthy individuals [75]. This report also demonstrated that an increase in *Veillonella* is positively correlated with IL-1β and IL-8 levels and the relative abundance of *Lactobacillales* in feces [75].

In a Chinese study of patients with PBC, *Acidobacteria*, *Lachnobacterium* sp., *Bacteroides eggerthii,* and *Ruminococcus bromii*, which are considered to be potentially beneficial bacteria, were reported to be decreased as compared with healthy controls, whereas pathogenic bacteria including *Enterobacteriaceae*, *Neisseriaceae*, *Veillonella*, *Actinobacillus pleuropneumoniae,* and *Haemophilus parainfluenzae* were increased in the feces of these patients [76]. Another study of PBC patients using 16S rRNA sequence analyses derived from fecal microbiota indicated a decrease in *Bacteroidetes* and an increase in *Fusobacteria* and *Proteobacteria* spp. [77]. Our recent study using the 16S rDNA sequence also revealed that bacterial diversity was lower in PBC patients with a decreased abundance of *Clostridium* and an increase in *Lactobacillus* [78]. As ursodeoxycholic acid (UDCA) is a common medication for PBC, we also investigated the difference in the microbial alteration between UDCA responders and nonresponders. We found a smaller population of the genus *Faecalibacterium* in the UDCA nonresponder group as compared with the UDCA responder group [78].

Autoimmune hepatitis (AIH) is another major autoimmune liver disease, but the pathogenesis of AIH still remains unclear. The first clinical study from Lin et al. revealed that the decreased abundance of *Bifidobacteria* and *Lactobacillus* is involved in the development of AIH, along with increasing plasma LPS levels in the later stage of AIH [79]. A recent study reported that as compared with healthy controls, AIH patients had a reduced diversity of intestinal microbiota and a change in bacteria species including *Streptococcus*, *Veillonella*, *Klebsiella*, and *Lactobacillus* [80]. Another study also reported that AIH patients demonstrated augmented intestinal permeability and bacterial translocation, which were associated with the severity of AIH activity, but not the stage of fibrosis [81].

In the experimental AIH model, germ-free mice demonstrated a much smaller amount of inflammatory cytokines and chemokines, including TNF-α, IL-4, interferon-γ, monocyte chemotactic protein-1, C-X-C motif chemokine 1, granulocyte colony-stimulating factor, and eotaxin. Concanavalin A (ConA)-induced apoptosis of liver cells was also significantly prevented in germ-free BALB/c mice as compared with positive controls [82]. Moreover, in conventional mice with ConA injection, plasma LPS levels were significantly higher than those in germ-free mice with ConA [82]. These data might explain the crucial role of LPS in ConA-induced liver injury in the mimicry of the pathogenesis of AIH.

### 3.5. Liver Cirrhosis and Its Complications

Liver cirrhosis is the end stage (F4) of liver fibrosis, which is characterized by the conversion of the normal liver architecture and regenerative nodules surrounded by tissue fibrosis and liver dysfunction [83]. Its complications, including ascites, varices, spontaneous bacterial peritonitis (SBP), and HCC, often require management during the clinical course, as they can result in death. As patients with liver cirrhosis are basically in an immunosuppressive state, infections can increase the rate of morbidity and mortality in cirrhotic patients [84]. An increase in the incidence of systemic endotoxemia in patients with liver cirrhosis has already been demonstrated using the Limulus amebocyte lysate test [85]. Likewise, subsequent quantitative endotoxin assays have revealed an elevation in plasma endotoxin levels along with liver fibrosis development to liver cirrhosis [86,87]. Another study reported the endotoxin levels in the portal blood flow of cirrhotic patients, suggesting that systemic endotoxemia is caused by the increased translocation of endotoxins from the gut via gut barrier dysfunction [88]. Taking into consideration the accumulating evidence, endotoxemia via leaky gut appears to be closely associated not only with SBP and liver fibrosis but also with the hyperdynamic state as well as renal, cardiac, pulmonary, and coagulation disturbances [1]. The disruption of the gut barrier is considered to be a pathological factor that can exaggerate liver cirrhosis and its complications [89]. Increased intestinal permeability is reported to be associated with the Child–Pugh classification, which indicates the stage of liver cirrhosis [90,91,92]. Thus, it is important to investigate the change in gut microbial diversity because of the key roles of bacterial translocation and endotoxemia in the pathogenesis of complications resulting from liver cirrhosis [93].

#### 3.5.1. Ascites and SBP

SBP is one of the most important infections in cirrhotic patients with ascitic fluid, which might be caused by bacterial translocation from the intestine. It has been reported that enteric Gram-negative bacteria, mainly the *Enterobacteriaceae* family, comprise the vast majority of causative microorganisms of SBP [94]. The impairment of small intestinal motility and SIBO are frequently observed in patients with liver cirrhosis with a history of SBP [95]. Bacterial DNA has also been identified as a factor associated with proinflammatory cytokine levels in ascites [96]. The presence of bacterial DNA in ascites and blood was observed in one third of cirrhotic patients with culture-negative ascites, which could worsen intrahepatic endothelial function and peripheral vasodilation [97,98]. Proton pump inhibitors (PPIs) are potent inducers of increased bacterial translocation, which might be associated with the promotion of intestinal overgrowth. Cirrhotic patients who use PPIs have demonstrated an increase in the prevalence of SBP [99]. The evidence from these studies shows that bacterial overgrowth could increase the risk of SBP via bacterial translocation resulting from intestinal hyperpermeability [100,101].

#### 3.5.2. Portal Hypertension

Along with the development of liver fibrosis, the portal blood pressure is elevated in patients with CLD as a result of the elevation in liver stiffness. As mentioned above, cirrhotic patients have an impaired gut barrier, resulting in bacterial translocation. An experimental study demonstrated that intraperitoneal injection of LPS can elevate the portal pressure and variceal bleeding, and increased portal hypertension can increase intestinal permeability [102,103,104]. Bacterial translocation induces the production of not only proinflammatory cytokines but also nitric oxide, endothelin, and cyclooxygenases [105,106,107]. These products also impair the contractility of the mesenteric vessels, which further exaggerates the elevation in the portal blood pressure [108,109].

Clinically, several studies, including our current study, have demonstrated that cirrhotic patients experience an elevation in plasma endotoxin levels and increased intestinal permeability after variceal hemorrhage [110,111]. Moreover, the results of a multivariate statistical analysis revealed that excessive intestinal permeability was an independent risk factor for infections such as SBP [110,112]. In these patients, poorly unabsorbable antibiotics including rifaximin could be useful for preventing SBP and the re-rupture of gastrointestinal varices [113,114]. The accumulating evidence suggests that there is a malignant circulation that induces portal hypertension via LPS stimulation through the gut–liver axis.

#### 3.5.3. Hepatic Encephalopathy

Hepatic encephalopathy (HE) is a major complication of end-stage liver cirrhosis, which induces symptoms due to an abnormal elevation in plasma ammonia levels, leading to confusion, lethargy, sleep disturbances, and coma [115]. An experimental animal model using germ-free mice in the mimicry of liver cirrhosis displayed a lower concentration of serum ammonia and reduced the level of neuroinflammation as compared with conventional mice [116]. This study also showed that the enrichment of *Lactobacillae* is positively associated with increased neuroinflammation in conventional mice with liver cirrhosis. In addition, LPS permeabilizes the blood–brain barrier in the same manner as the gut barrier and affects the brain microglia through the production of nitric oxide, which induces the swelling of astrocytes in patients with HE [117,118]. Likewise, an association was reported between endotoxemia and minimal HE and an increase in the incidence of overt HE [117].

There is no significant difference in the profiles of fecal microbiota between cirrhotic patients with HE and those without HE [12,119]. However, the diversity of *Alcaligeneceae* in HE patients was shown to be increased as compared with normal controls. *Alcaligeneceae* is reported to contribute to the production of ammonia, and this change was observed only among most HE patients taking PPIs. In contrast, none of the healthy controls taking PPIs demonstrated an increase in this alteration [12]. It has also been demonstrated that SIBO is closely correlated with minimal HE [120]. Bajaj et al. reported a higher abundance of fecal *Enterobacteriaceae*, which is a major bacterium that produces a large amount of endotoxins [121].

### 3.6. HCC

Changes in the gut microbiota affect the hepatic tumor environment. Hepatic inflammation caused by leaky gut can drive tumorigenesis in the liver. The results of several animal experiments have suggested that LPS–TLR4 signaling can induce the promotion of HCC development [122,123,124]. It has also been reported that HCC model mice treated with antibiotics and germ-free mice demonstrated less carcinogenesis. In this study, TLR4 signaling in the liver cells was partially mediated by the inhibition of hepatocyte apoptosis and growth signals, including epiregulin derived from HSCs [124]. Dysbiosis-induced bacterial metabolites, such as deoxycholic acid, are additional factors that promote hepatocarcinogenesis related to gut-derived endotoxemia, which could enhance liver inflammation via TLR pathways [125].

In clinical studies, a higher abundance of intestinal *E. coli* was found to be associated with the presence of HCC and reported to produce endotoxins [20]. Animal studies have also indicated that endotoxins play an important role in the progression of HCC [126]. These studies further suggested that an increase in fecal *Bacteroides* is probably accompanied by an elevation in proinflammatory cytokine levels such as IL-8 and IL-13, activated monocytes, and monocytic myeloid-derived suppressor cells, which play a role in the hepatocarcinogenesis of NAFLD patients [7]. *Klebsiella-* and *Haemophilus*-producing LPS are also reported to be increased in patients with HCC in the early stage as compared with non-HCC controls [19,127,128]. This growing evidence may suggest the possibility of a noninvasive predictive marker for the detection of HCC incidence in the future.

## 4. Conclusions

Various studies have revealed that gut-derived endotoxins from the gut microbiota could be closely associated with CLDs. Lifestyle factors, such as dietary factors and alcohol drinking, are fundamental therapeutic targets for preventing the pathogenesis of liver diseases in terms of the gut–liver axis. Dysbiosis and leaky gut are induced by alcohol and a high-calorie/high-fat diet, which results in liver inflammation and liver fibrosis progression, eventually progressing to liver cirrhosis, especially in patients with alcoholic liver disease and NAFLD. As mentioned above, it is already known that endotoxemia is associated with the occurrence of cirrhotic complications in patients with liver cirrhosis via activation of TLR4 signaling in KCs and HSCs. This gut–liver axis resulting from gut dysbiosis might also induce the development of HCC. Considering these findings, the gut microbiota and gut-derived endotoxemia could be useful therapeutic targets in the management of the progression of liver pathogenesis, and further investigation in this field is expected in the future.

## Figures and Tables

**Figure 1 ijms-22-06921-f001:**
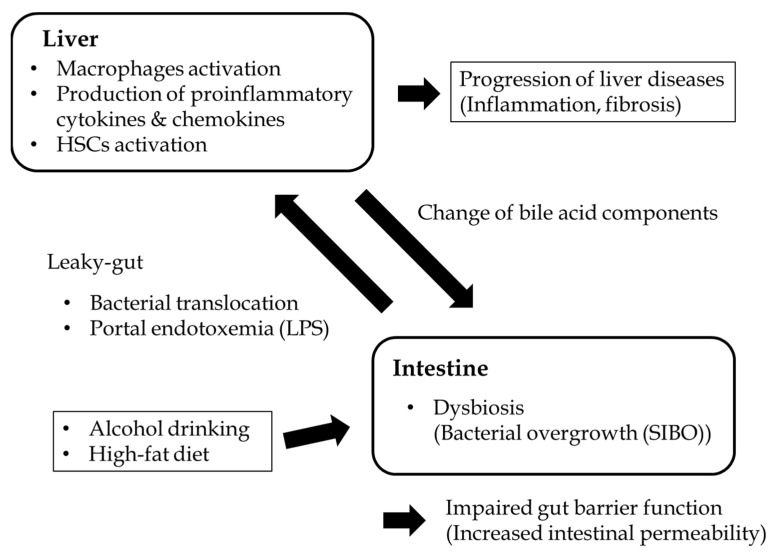
Schema of the interaction between the gut and liver in CLDs. Lifestyle factors such as drinking alcohol and consuming a high-fat diet alter the diversity of the gut microbiota and result in bacterial overgrowth, increasing bacterial translocation including endotoxins (lipopolysaccharides; LPS). LPS can translocate into the portal vein via the disruption of the gut barrier, eventually reaching into the liver as a result of increased intestinal permeability, strongly activating regional liver immune cells, mainly Kupffer cells, which produce proinflammatory cytokines and chemokines. LPS also stimulate the activation of hepatic stellate cells, which play a main role in the progression of liver fibrosis via the production of extracellular matrix proteins such as collagens.

**Figure 2 ijms-22-06921-f002:**
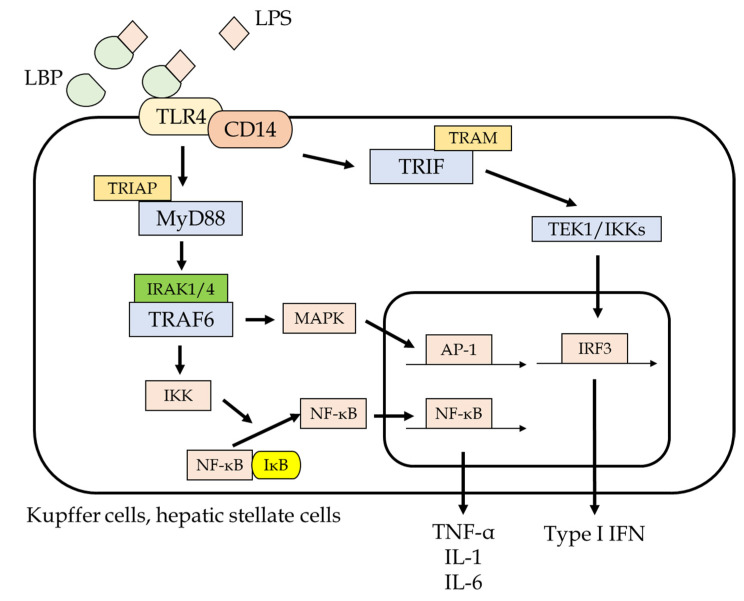
Intracellular signaling of the TLR4 pathway via LPS binding. Liver cells such as Kupffer cells and hepatic stellate cells contain TLR4, which mediates the innate immunity response. LPS binding to LPS binding protein, which is a translocator for LPS, attaches to TLR4 and stimulates its downstream signal, including NF-κB and mitogen-activated protein kinase, which induces the transcription of proinflammatory cytokines and chemokines such as TNF-α, IL-1, and IL-6. These inflammatory cytokines cause liver inflammation as well as activating hepatic stellate cells, leading to excessive production of the extracellular matrix, which suggests that chronic exposure of LPS can result in the progression of liver fibrosis, culminating in liver cirrhosis.

**Table 1 ijms-22-06921-t001:** Representative changes in microbial diversity according to types of CLDs.

Type of Diseases	Bacteria				
	Phylum	Class	Order	Family	Genus (Species)
**ALD**	Firmicutes Bacteroidetes ↓ Proteobacteria ↑	Bacilli Clostridia Negativicutes Bacteroidia proteobacteria ↑	Lactobacilales Clostridiales Selenomonadales Bacteroidales Enterobacteriales	Lactobacillaceae Clostridiaceae Ruminococcaceae ↓ Lachnospiraceae Veillonellaceae ↑ Bacteroidaceae ↓ Porphyromonadaceae ↓ Enterobacteriaceae ↑	Lactobacillus ↓ Clostridium ↓ Faecalibacterium ↓ Dorea ↑ Blautia ↑ Klebsiella ↑
**NAFLD**	Firmicutes Bacteroidetes ↑↓ Proteobacteria ↑	Bacilli Clostridia ↓ Bacteroidia proteobacteria ↑	Lactobacilales Clostridiales ↓ Bacteroidales Enterobacteriales	Lactobacillaceae ↑ Clostridiaceae Ruminococcaceae ↓ Lachnospiraceae Bacteroidaceae ↓ Porphyromonadaceae ↑↓ Enterobacteriaceae ↑	Lactobacillus ↑ Clostridium Faecalibacterium ↓ Dorea ↑ Bacteroides ↓ Escherichia ↑
**HBV**	Firmicutes Bacteroidetes ↓ Proteobacteria ↑ Actinobacteria	Bacilli Clostridia Bacteroidia Proteobacteria Betaproteobacteria Actinobacteriia	Lactobacilales Clostridiales Bacteroidales Enterobacteriales Neisseriales Bifidobacteriales	Lactobacillaceae ↓ Enterococcaceae ↑ Ruminococcaceae Bacteroidaceae Enterobacteriaceae ↑ Neisseriaceae ↑ Bifidobacteriaceae ↓	Lactobacillus ↓ Enterococcus ↑ Faecalibacterium Bacteroides Bifidobacterium ↓
**HCV**	Firmicutes ↓ Bacteroidetes ↑ Proteobacteria ↑ Actinobacteria	Bacilli Bacteroidia Proteobacteria Actinobacteriia	Lactobacilales Bacillales Enterobacteriales Bifidobacteriales	Lactobacillaceae Enterococcaceae ↑ Stphylococcaceae Enterobacteriaceae ↑ Bifidobacteriaceae	Lactobacillus Enterococcus ↑ Staphylococcus ↑ Bifidobacterium
**PBC**	Firmicutes Bacteroidetes ↑↓ Proteobacteria ↑ Fusobacteria Acidobacteria ↓	Bacilli Clostridia Negativicutes Proteobacteria Gammaproteobacteria Fusobacteriia	Lactobacilales Clostridiales Selenomonadales Enterobacteriales Pasteurellales Fusobacteriales	Lactobacillaceae Clostridiaceae Ruminococcaceae Lachnospiraceae Veillonellaceae Enterobacteriaceae ↑ Pasteurellaceae Fusobacteriaceae	Lactobacillus ↑ Eubacterium ↑ Clostridium ↓ Ruminococcus ↓ Lachnobacterium ↓ Veillonella ↑ Actinobacillus ↑ Haemophilus ↑ Fusobacterium ↑↓
**AIH**	Firmicutes Actinobacteria Bacteroidetes Proteobacteria	Bacilli Actinobacteriia Negativicutes Proteobacteria	Lactobacilales Bifidobacteriales Selenomonadales Enterobacteriales	Lactobacillaceae Bifidobacteriaceae Veillonellaceae Streptococcaceae Enterobacteriaceae	Lactobacillus ↑↓ Bifidobacterium ↓ Veillonella ↑ Streptococcus ↑ Klebsiella ↑

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
