# Peer review of "Intestinal Permeability Is a Mechanical Rheostat in the Pathogenesis of Liver Cirrhosis"

_ijms, 2021, doi:10.3390/ijms22136921_

Round 1

Reviewer 1 Report

Gut microbiota are important for homeostasis of mammals. Gut dysbiosis caused by a high –fat diet and alcohol consumption have been correlated with liver diseases. This is a concise review summarizing changes of gut microbiota and liver diseases. Before publications, the reviewer suggests the authors consider the following points.

  1. In page 2, the authors describe “dysbiosis including SIBO, “ whereas dybiosis and SIBO are in parallel in line 15 of page 7, in line 10 of page 10, and in Figure 1. Please keep consistency.

  1. Similarly, in line 15 of page 7, the authors suggest dysbiosis causes increased intestinal permeability. On the other hand, leaky gut and dysbiosis are in parallel in the section “HCC” in page 18.

  1. The reviewer assumes “correction (or improvement) of dysbiosis could contribute to preventing the disruption of but barrier functions” in the last part of page 7.

  1. In line 4 of page 10, “Our study ,,,,,,,” is supported by a previous work of the authors? Quote a reference or add “unpublished data”.

  1. The authors explain the different reactivity of germ free mice and conventional mice against ConA treatment in page 14. Please check conjunctions “In contrast” and “Moreover” are appropriate in this context.

  1. It is not clear what “them” indicates in line 4 of page 6. The reviewer assumes that the sentence may be “which connect claudins (or other TJ molecules) to the actin cytoskeleton”.

  1. A table summarizing types of liver injury and affected bacteria spices may be helpful to understand the correlation between dysbiosis and liver diseases.

  1. In abstract line 16, add “alpha” after “tumor necrosis factor”

Author Response

Thank you so much for reviewing our manuscript and your sincerely suggestion. I have corrected each point as below.

1) In page 2, the authors describe “dysbiosis including SIBO“, whereas dysbiosis and SIBO are in parallel in line 15 of page 7, in line 10 of page 10, and in Figure 1. Please keep consistency.

Response:Thank you for your reviewing. I have deleted SIBO in page 2 and other parts and arranged Figure 1. In the paragraph of SIBO, as I explained SIBO is a part of dysbiosis, I don’t think it would be necessary to address both of dysbiosis and SIBO.

2) Similarly, in line 15 of page 7, the authors suggest dysbiosis causes increased intestinal permeability. On the other hand, leaky gut and dysbiosis are in parallel in the section “HCC” in page 18.

Response: Thank you so much for your comment. I have deleted dysbiosis from the part in page 18.

3) The reviewer assumes “correction (or improvement) of dysbiosis could contribute to preventing the disruption of but barrier functions” in the last part of page 7.

Response: I have corrected it to the appropriate sentence like “This finding could corroborate that improvement of dysbiosis contributes to preventing the disruption of gut–barrier function”.

4) In line 4 of page 10, “Our study ,,,,,,,” is supported by a previous work of the authors? Quote a reference or add “unpublished data”.

Response: Thank you so much for your suggestion. I have added the reference for our previous work already published in that part. In addition, as I realized the numbers of references after this were inappropriate, I have corrected their numbering too.

5) The authors explain the different reactivity of germ-free mice and conventional mice against ConA treatment in page 14. Please check conjunctions “In contrast” and “Moreover” are appropriate in this context.

Response: I agree with the reviewer. I have corrected these conjunctions. Please see the part in page 14.

6) It is not clear what “them” indicates in line 4 of page 6. The reviewer assumes that the sentence may be “which connect claudins (or other TJ molecules) to the actin cytoskeleton”.

Response: I agree with the reviewer’s comment. I have rearranged the sentence as below.

“These proteins bind scaffolding proteins such as zonula occludins 1 (ZO-1), ZO-2, and ZO-3, which connect transmembrane proteins including claudins and occludins to the actin cytoskeleton.”

7) A table summarizing types of liver injury and affected bacteria spices may be helpful to understand the correlation between dysbiosis and liver diseases.

Response: Thank you so much for your suggestion. I have made a table for a summary of bacteria described in this review according to types of liver diseases. Please see the last page of the revised manuscript.

8) In abstract line 16, add “alpha” after “tumor necrosis factor”

Response: I have added alpha in that part.

Reviewer 2 Report

The Authors of the manuscript aimed to analyze the role of gut-derived endotoxemia underlying the progression of liver pathogenesis. They analyzed the mechanisms of endotoxemia derived from gut microbiota (i.e. gut dysbiosis, intestinal bacterial overgrowth, disruptions in intestinal barrier function) and  the role of endotoxemia on the progression of liver pathogenesis. This review paper is a significant contribution to the scientific discussion about role of gut dysbiosis in the pathogenesis of liver diseases. It has good scientific quality. The manuscript can be accepted with only minor editorial and linguistic corrections.

Author Response

The Authors of the manuscript aimed to analyze the role of gut-derived endotoxemia underlying the progression of liver pathogenesis. They analyzed the mechanisms of endotoxemia derived from gut microbiota (i.e. gut dysbiosis, intestinal bacterial overgrowth, disruptions in intestinal barrier function) and the role of endotoxemia on the progression of liver pathogenesis. This review paper is a significant contribution to the scientific discussion about role of gut dysbiosis in the pathogenesis of liver diseases. It has good scientific quality. The manuscript can be accepted with only minor editorial and linguistic corrections.

Response: Thank you so much for reviewing our manuscript. We have corrected spelling errors and grammars again. The manuscript has also been checked by native English speakers.